# Detonation synthesis of carbon nano-onions via liquid carbon condensation

M. Bagge-Hansen [1], S. Bastea [1], J.A. Hammons [1], M.H. Nielsen [1], L.M. Lauderbach[1], R.L. Hodgin[1], P. Pagoria [1], C. May[1], S. Aloni[2], A. Jones[1], W.L. Shaw [1], E.V. Bukovsky [1], N. Sinclair[3], R.L. Gustavsen[4], E.B. Watkins[4], B.J. Jensen[4], D.M. Dattelbaum[4], M.A. Firestone [4], R.C. Huber[4], B.S. Ringstrand[4], J.R.I. Lee [1], T. van Buuren[1], L.E. Fried [1] & T.M. Willey [1]

Transit through the carbon liquid phase has significant consequences for the subsequent formation of solid nanocarbon detonation products. We report dynamic measurements of liquid carbon condensation and solidification into nano-onions over ∽200 ns by analysis of time-resolved, small-angle X-ray scattering data acquired during detonation of a hydrogen-free explosive, DNTF (3,4-bis(3-nitrofurazan-4-yl)furoxan). Further, thermochemical modeling predicts a direct liquid to solid graphite phase transition for DNTF products ~200 ns post-detonation. Solid detonation products were collected and characterized by high-resolution electron microscopy to confirm the abundance of carbon nano-onions with an average diameter of ∽10 nm, matching the dynamic measurements. We analyze other carbon-rich explosives by similar methods to systematically explore different regions of the carbon phase diagram traversed during detonation. Our results suggest a potential pathway to the efficient production of carbon nano-onions, while offering insight into the phase transformation kinetics of liquid carbon under extreme pressures and temperatures.

---

[1] Lawrence Livermore National Laboratory, 7000 East Ave., Livermore, CA 94550, USA. [2] The Molecular Foundry, Lawrence Berkeley National Laboratory, 67 Cyclotron Rd., Berkeley, CA 94720, USA. [3] Dynamic Compression Sector (DCS), Institute for Shock Physics, Washington State University, 9700S. Cass Ave., Argonne, IL 60439, USA. [4] Los Alamos National Laboratory, Bikini Atoll Rd., SM 30, Los Alamos, NM 87545, USA. Correspondence and requests for materials should be addressed to T.M.W. (email: willey1@llnl.gov)

Solid carbon products, such as nanodiamond, are produced under the high pressures and temperatures (>10 GPa, > 3000 K) generated during the detonation of many common high explosives (HEs)[1–3]. This observation, made as early as 1963[4], has subsequently been exploited for the mass production and broad technical and industrial application of nanodiamond, yet relatively little is certain about how and when these products form during the detonation sequence[1,2]. By understanding these processes, the technological impact of detonation nanodiamond could be mirrored in other pure and doped nanocarbon materials. More generally, revealing the kinetics of carbon condensation in HE products has profound technological and practical implications, e.g., in enabling more efficient production of other potentially useful carbon nanoallotropes, predicting the energy release behavior, identifying failure mechanisms, and decreasing sensitivity of HEs;[5] nevertheless, nanocarbon formation and evolution during detonation remain experimentally underexplored because the violent, dynamic, and opaque nature of detonations of even small quantities of HE material are extremely challenging to interrogate—especially with the nanosecond-scale temporal resolution required to monitor product formation kinetics.

Thermochemical modeling of HE detonations has matured somewhat faster than complementary experimental techniques and offers instructive, albeit incomplete, insights into the expected condensation products[6–13]. Many HEs are relatively simple molecular solids composed of carbon, hydrogen, nitrogen, and oxygen, where a negative oxygen balance leads to excess carbon in the detonation products. During detonation, the product species, which are a complex mixture of molecular gases (e.g., $N_2$, $H_2O$, $CO$, $CO_2$, etc.), ionic species (e.g., $OH^-$, $H^+$, etc.), and carbon condensates, evolve and reach full chemical equilibrium at the Chapman–Jouguet (C–J) point[11,12,14], then cool through adiabatic expansion over several microseconds. The thermodynamic properties of carbon along this expansion path (specific to the HE) are inferred by comparison to the carbon phase diagram; however, the size of the condensates is an important factor since the surface energy of nanoscale carbon clusters is a significant term in determining the chemical potential and leads to substantial differences from bulk behavior[15]. The effects of initiation defects or hot spots, which likely produce higher initial local temperatures, are negligible as they represent a minor volume fraction of the subsequent products.

Only a few HEs, including 3,4-bis(3-nitrofurazan-4-yl)furoxan (DNTF) and benzotrifuroxan (BTF), are predicted to reach detonation pressures and temperatures compatible with stable liquid carbon condensates at early times, but no conclusive experimental evidence is currently available. Nonetheless, evidence of the liquid phase is critical for elucidating the location of the carbon liquidus for carbon nanoparticles and may open new avenues for the detonation synthesis of carbon-based materials.

The liquid phase of carbon is very difficult to study experimentally and has been observed above the graphite/diamond/liquid triple point at ~12 GPa/5000 K using laser or Joule heating of bulk carbon[16,17]. In detonation literature, however, the liquid state is usually inferred from the size and morphology of nanodiamonds in late time detonation products, notably for BTF[18,19]. From these indirect observations, liquid droplet formation and solidification have been suggested to occur over several microseconds through coalescence of smaller droplets and concomitant adiabatic cooling; however, no early time experimental results have verified the existence of liquid carbon or the timescales suggested during HE detonation[2]. Moreover, these relatively long timescales are generally inconsistent with thermochemical models in which the liquid state vanishes within one microsecond.

Herein, we combine our thermochemical modeling predictions with time-resolved, small-angle X-ray scattering (TR-SAXS) over the first microsecond post-detonation, and late-time product collection and microscopy to build a consistent picture of carbon condensation and the resulting nano-carbon products from HEs that reach different areas of the carbon phase diagram upon detonation, including liquid carbon for DNTF. Significantly, we report the direct observation of a transient liquid carbon phase in the detonation products of DNTF, followed by prompt solidification into stable, abundant nano-onions.

## Results

**Thermochemical modeling**. Modern thermochemical calculations suggest that the conditions reached by different HEs during detonation may be in different regions of the carbon phase diagram depending on e.g., composition, chemical energy content, and initial density. Figure 1a illustrates this point—through thermochemical modeling (see Methods) for three materials: hexanitrostilbene (HNS – $C_{14}H_6N_6O_{12}$); 60% cyclotrimethylenetrinitramine (RDX – $C_3H_6N_6O_6$)/40% 2,4,6-trinitrotoluene (TNT – $C_7H_5N_3O_6$) by mass (Composition B or Comp B); and DNTF ($C_6N_8O_8$). The hydrogen-free DNTF[20] has the highest predicted detonation temperature, due to the absence of water as a detonation product. Further, this selection of HEs include detonation conditions that span every region of the carbon phase diagram; notably, this includes the liquid phase for DNTF.

**Time-resolved X-ray measurements and SAXS modeling**. Our experimental SAXS results provide a systematic study of carbon condensation kinetics with temporal resolution better than 50 ns[21]. During detonation of pressed pellets of HE, we collect synchrotron-based, TR-SAXS from discrete X-ray pulses, every 153.4 ns (Fig. 1b, c). A nominally identical series of shots is interleaved to provide a time series from detonation to steady-state. A complete description of the experiment, data collection, and analysis can be found in the Methods section.

The TR-SAXS patterns collected for DNTF are shown in Fig. 2a and Supplementary Figs. 5 and 6. For times <~65 ns post-denotation, assuming a constant DNTF detonation front velocity, we expect a superposition of detonation products and undetonated DNTF from transmission of the primary X-ray beam through the detonation front curvature (Supplementary Fig. 9), across the diameter of the cylindrical pellet (Fig. 1b); in particular, the undetonated DNTF manifests as scattering intensity at low-$q$ (Fig. 2a), associated with micron-scale features (e.g., voids) in the pressed pellet[22,23]. These data provide an internal timing confirmation for our subsequent analysis. In concert, at early times (<120 ns), we observe the scattering from emerging denotation products. Surprisingly, the high-$q$ ($q > 6 \times 10^{-2} Å^{-1}$) region of these scattering data can be fully explained using a simple Porod approximation—i.e., a homogenous phase with a smooth, well-defined surface[22]; as in all SAXS data, a flat background is also present and dominates at $q > 0.2 Å^{-1}$. A good fit to the Porod approximation for these data suggests that the carbon condensates at these early times are smooth 3D particles—e.g., liquid droplets. At times greater than 120 ns, additional intensity from small-scale heterogeneities appears at high-$q$ ($6 \times 10^{-2}$ to $0.2 Å^{-1}$) that is not captured by the Porod approximation; further, we observe distortion in the low-$q$ region that is characteristic of more than one length scale (e.g., heterogeneous phase, anisotropic shape or multi-modal distributions). Therefore, a more complex model is required to fit the results at $t > 120$ ns.

In order to capture a potential carbon phase change from liquid to nano-onion, we require a scattering model that describes the entire $q$-region measured in these TR-SAXS experiments. The scattering model for both a liquid droplet and a nano-onion are

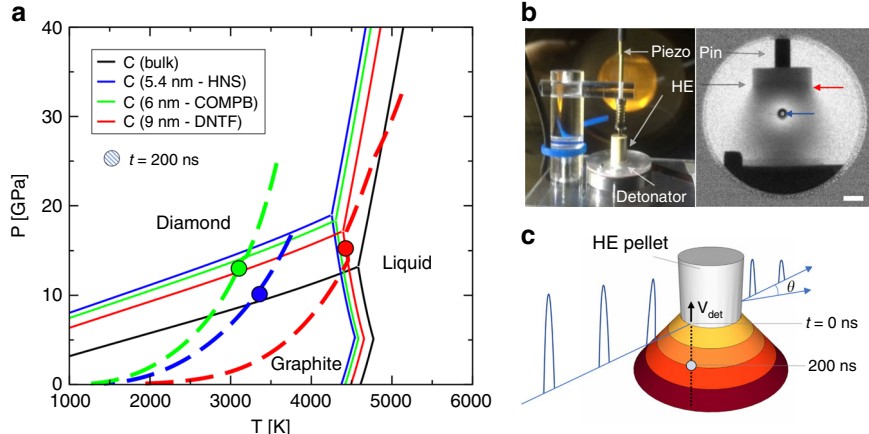

**Fig. 1** Carbon phase diagram and TR-SAXS method. **a** Adiabats for HNS, Comp B, and DNTF (blue, green, and red dashed lines, respectively) calculated using thermochemical modeling. Solid circles along these adiabats identify the calculated position for the detonation products at $t = 200$ ns post-detonation. These data are superimposed on the computed carbon phase diagram with explicit consideration for particle size, calculated for sizes of 5.4 nm, 6 nm, 9 nm, and bulk. Phase boundaries for these sizes are delineated by blue, green, red, and black solid lines, respectively. Diamond forms at pressures and temperatures in the upper left, metastable graphite at the lower left, and liquid at the far right. In the high explosive DNTF, ~200 ns marks a phase transition between liquid and solid graphitic carbon products. **b** A small-scale detonation assembly typical for synchrotron, time-resolved small-angle X-ray scattering experiments and dynamic scatter-beam imaging during detonation. Note the curved detonation front (red arrow) at the interface between the undetonated HE (e.g., HNS shown) and the denotation products, which include carbon condensates; also, the relative position of the X-ray beam is apparent from the location of the primary beam stop (blue arrow). Scale bar is 2 mm. **c** Schematic snapshot of the denotation front propagating with velocity, $V_{det}$, along the axis of the pellet subsequent to initiation at the base of the pellet. The interface between the detonation front and the HE, at the elevation of the coincident X-ray beam (blue arrow), is defined as $t = 0$ ns. The evolution of detonation products occurs over several hundred nanoseconds post-detonation, in concert with adiabatic cooling

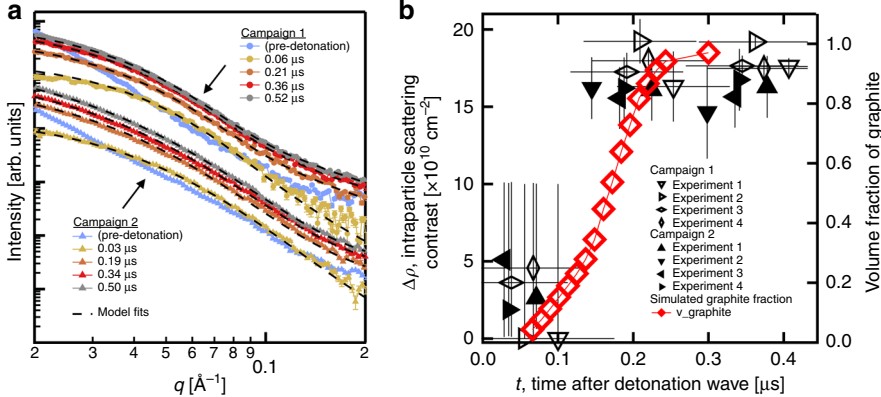

**Fig. 2** DNTF TR-SAXS data and modeling. **a** TR-SAXS acquired during detonation of the high explosive DNTF. Two separate experimental campaigns conducted several years apart demonstrate repeatability in the SAXS time evolution, particularly over the first 300 ns after detonation. Data shown are from one representative shot from each campaign; further data can be found in Supplementary Figs. 5 and 6. Error bars are calculated using the standard error of mean (Methods). **b** The subtle and systematic changes seen in the TR-SAXS profiles are further analyzed through SAXS modeling (Supplementary Methods). Here, a scattering contrast parameter from the model, $\Delta\rho$, captures the transition from a homogenous solid at $t < 0.2$ μs to a solid with heterogeneity emerging for t > 0.2 μs. This heterogeneity is generated by the formation of carbon nano-onions from homogeneous liquid nano-droplets. Further, a thermochemical model of DNTF detonation products was employed in hydrodynamic simulations to calculate the fraction of graphite, $\nu_{graphite}$ (i.e., nano-onions) as a function of time post-detonation, as seen by the TR-SAXS beam traversing the expanding products; at each time shown, the remainder of the carbon volume is liquid, indicating a phase transition at t~0.2 μs. This comparison reveals profound agreement between the predicted liquid to nano-onion phase transition and the TR-SAXS. Temporal uncertainty is ± 75 ns in campaign 1 and ± 5 ns in campaign 2 (Methods). Error bars for $\Delta\rho$ are calculated using the standard error (Supplementary Methods). Source data are provided as a Source Data file

derived from the well-known scattering equation for any spherically symmetric phase:[23]

$$I(q) = \left[ 4\pi \int_0^R \rho(r) \frac{\sin qr}{qr} r^2 dr \right]^2 \quad (1)$$

where $R$ is the radius of the phase and $\rho(r)$ describes the radial scattering length density. As a spherical liquid droplet freezes into

a carbon nano-onion, concentric graphitic layers form. This transition can be modeled by using periodic function (e.g., a rectangular wave) for $\rho(r)$, which approximates the graphitic layers within the nano-onion (Supplementary Fig. 2) and allows for a straightforward analytical solution. The interlayer spacing between concentric spherical shells is set to 3.1 Å, based on transmission electron microscopy (TEM) results, below. Therefore, Eq. 1 provides the foundation for formulating the scattering model so that the interlayer contrast, $\Delta\rho$, can be obtained from each

TR-SAXS trace using as few fit parameters as possible (Supplementary Eqs. 1–2). In this particular system, $\Delta\rho$ has a non-linear effect on the TR-SAXS model, whereby high-$q$ heterogeneities are not observed until $\Delta\rho$ reaches $\sim 15 \times 10^{10}$ cm$^{-2}$ (Supplementary Fig. 3). Consequently, while the model can readily determine when onion-like scattering is observed, it fails to provide information about potential transition states. Further, Eq. 1 is easily incorporated into a comprehensive scattering model that accounts for the size-distribution and volume fraction of the nano-onions, as well as the flat background (Supplementary Eqs. 3–6, Supplementary Tables 1 and 2). Plotting $\Delta\rho$ as a function of time (Fig. 2b) illustrates that the interlayer scattering contrast increases most significantly at $\sim 200$ ns post-detonation, indicating a potential solidification of liquid carbon droplets into solid nano-onions. It is notable that the data from many detonations across two distinct experimental campaigns with DNTF show strong agreement. Further, we superimpose the computed liquid to graphite transition for DNTF (Fig. 2b) directly from hydrodynamic/thermochemical simulations (Fig. 1a). The coincidence of these experimental and simulated results is remarkable; both reveal a clear change between 150 and 200 ns. The TR-SAXS analysis additionally provides the corresponding size distribution (Supplementary Fig. 4) which suggests that the mode diameter evolves from $\sim 9$ nm at early times to $\sim 10$ nm after 200 ns, concurrent with the proposed phase change. This expansion is consistent with the volume expansion anticipated during solidification from liquid to graphitic carbon[15].

With the goal of providing a comprehensive picture of carbon production in detonation events, we also performed TR-SAXS on detonating HNS and Comp B. The predicted detonation points for these materials are within the graphite and diamond regions, respectively, of the carbon phase diagram (Fig. 1a). In the $q$-range that was measured ($1.5 \times 10^{-2}$ to $2.5 \times 10^{-1}$ Å$^{-1}$), solid carbon products dominate the scattering. A direct comparison of the TR-SAXS profiles from HNS, Comp B, and DNTF at 500 ns post-detonation (Fig. 3) confirms that the carbon condensation products of these HEs display strong differences, even at early times. These differences are manifested in: (1) the position and breadth of the low-$q$ Guinier knee, and (2) the slope and shape of the intensity decay subsequent to the Guinier knee[22,23]. While the $q$-position of the Guinier knee indicates the size distribution of nanocarbon products, the intensity decay that follows depends on

small-scale scattering heterogeneities that connote the morphology and phase. In TR-SAXS of HNS, detonation products exhibit a radius of gyration of 3 nm and significant fine-structure (Supplementary Eq. 17, Supplementary Fig. 13) that suggest the formation of convoluted structures of quasi-2D carbon condensates with relatively small primary particle dimensions[24]. Conversely, the TR-SAXS from Comp B shows a broad Guinier knee that is followed by a much steeper intensity decay and faint Guinier knee at high-$q$, which is characteristic of detonation nanodiamonds with some surface texture[25]. A TR-SAXS pattern similar to that obtained from Comp B (Fig. 3) is not observed at any point during the detonation of DNTF and underscores that nanodiamond is not the dominant phase during the DNTF detonation. Modeling of the Comp B TR-SAXS data confirm that the formation of largely spherical nanodiamonds provides the best-fit (Supplementary Eqs. 13–16 Supplementary Fig. 12) and is in very good agreement with prior studies of detonation nanodiamonds[26–28]. With all the HEs measured, despite very different C–J points, the TR-SAXS data show consistent patterns emerge rapidly—within a few hundred ns after detonation—and thereafter, only the scaled intensity varies due to the dynamic scattering contrast of the expanding background gases[24,29,30]. After $\sim 500$ ns, for all HEs studied, we did not observe any further significant time-dependent changes in the shape of the TR-SAXS profiles, despite extending data collection out to several microseconds.

**Detonation Product Recovery and TEM Characterization.** Characterization of recovered detonation products offers valuable insight into the phase and morphology of solid carbon particles that cannot be gleaned from the TR-SAXS data alone. We designed a separate set of experiments to collect carbon products from small-scale detonations of DNTF, HNS, and Comp B for analysis by high-resolution TEM and electron energy loss spectroscopy (EELS). Our collection experiments were guided by two principles. First, the detonation should proceed under steady conditions and with no (or minimal) late-time re-shock of detonation products—which could substantially change the amount, phase, morphology, and distribution of nanocarbon produced behind the detonation shock front[31]. Second, we severely limited sources of carbon contamination and the

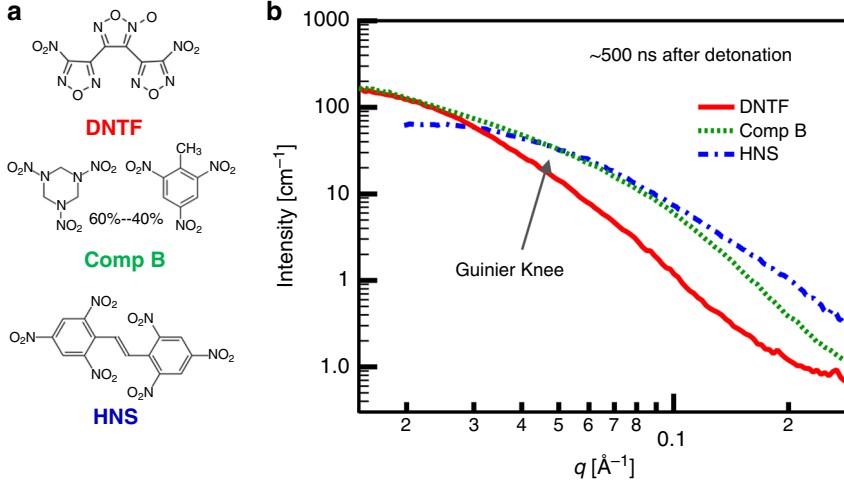

**Fig. 3** HE molecules and TR-SAXS data. **a** Molecular structures for three HE materials measured by TR-SAXS. **b** TR-SAXS profiles, acquired about 500 ns after detonation, for the three explosives exhibit dramatic variation associated with different nanocarbon products. In particular, the Guinier knee is associated with the size distribution of products; for DNTF (red solid line), the larger size distribution of nano-onions (average diameter ~10 nm) manifests in a Guinier knee at much lower $q$ compared to HNS (blue dash-dotted line) and Comp B (green dotted line). Data are provided in a Source Data file

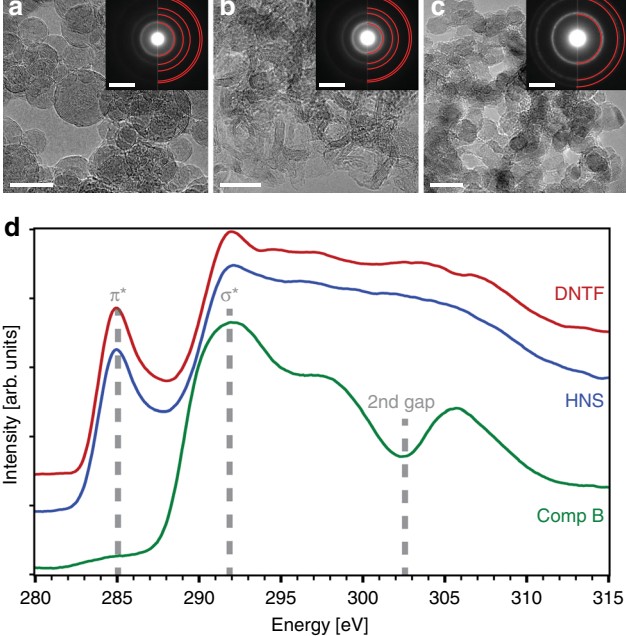

**Fig. 4** TEM analysis of nanocarbon products. **a–c** Transmission electron micrographs and diffraction data (insets) from recovered detonation products of **a** DNTF, **b** HNS, and **c** Comp B, with diffraction spacings for relevant carbon phases highlighted by red semi-circles. DNTF and HNS products largely consist of graphitic structures with onion-like and fiber-like morphologies, respectively, while Comp B contains a significant amount of diamond. Differences in the carbon allotropes for the three recovered products are also clear in carbon K-edge electron energy loss spectra **d**. Scale bars are 10 nm **a–c** and 5 nm$^{-1}$ (insets). Source data are provided as a Source Data file

possibility of the condensed products mixing with solid impurities. We achieved these requirements by capturing the detonation products in ice, as already demonstrated by others[4]. (See Methods for a description of the experimental set-up). We employ ice here primarily for soft capture of solid products, following limited expansion of detonating unconfined HE charges; experiments with lower-density, packed snow yield virtually indistinguishable results. Further, other than drying, the collected products were not subjected to any post-treatments (e.g., acid, high heat, rinsing, etc.). Figure 4 presents high-resolution TEM micrographs (Fig. 4a–c), corresponding electron diffraction patterns (Fig. 4a–c insets), and carbon K-edge EELS spectra (Fig. 4d) for DNTF, HNS, and Comp B. The observed carbon products are remarkably different from one another.

Mixtures of carbon allotropes (graphite, amorphous, and diamond) are observed in the solid products produced by HNS and Comp B. HNS products consist primarily of ribbons between 2–5 nm wide. Diffraction data and image FFT (Fig. 4b) show clear polycrystalline graphite features, and carbon K-edge EELS spectra (Fig. 4d) contain a strong π$^\star$ peak consistent with graphite. In Comp B, agglomerated 5–7 nm nanodiamonds are pervasive throughout, as confirmed by imaging, diffraction, and carbon K-edge EELS (Fig. 4c and d). No significant graphitic shells were observed on individual diamond nanoparticles within these agglomerates via TEM. Detonations of Comp B, particularly in ice, is well-known to produce a relative abundance of nanodiamond with morphologies consistent with these observations[4].

The solid detonation products of DNTF overwhelmingly contain carbon nano-onions—concentric graphitic layers extending inward from the spherical particle surface (Fig. 4a, Supplementary Fig. 1). No other carbon phase was consistently observed

despite rigorous TEM characterization of multiple samples (see Methods). These nano-onions have a broad size distribution (Supplementary Fig. 1) and an average diameter of ~10 nm. The nano-onions layers have (002) spacings that are compressed from the typical 3.35 Å observed for graphite. Measurements from the FFT of close to sixty images as well as over ten diffraction patterns of these nano-onions show (002) spacings ranging between 3.05 Å and 3.24 Å, with an average spacing of 3.15 Å. Compressed interlayer spacings in nano-onions have been seen before, with (002) spacings as small as 2.7 Å[32–34]; the mechanism that produces these compressed layers is a matter under further investigation. The carbon K-edge EELS spectra (e.g., Fig. 4d) display a distinct π$^\star$ peak consistent with $sp^2$ carbon and agree with the observation of carbon nano-onions. The size, morphology and structure of these late-time condensed products are remarkably consistent with those inferred from the TR-SAXS data at 500 ns (after which little change in the SAXS was observed).

## Discussion

The TEM analysis of DNTF detonation products constitutes the first observation of carbon nano-onion formation as a major, recoverable nanocarbon phase from a detonation event (Supplementary Fig. 1). The TR-SAXS data show an apparent transition for DNTF condensed products (Fig. 2) from only Porod-conforming scattering at early times to additional interlayer scattering at t > 200 ns that strongly suggests that these carbon nano-onions form by solidification from spherical liquid droplets. Indeed, this is in close agreement with the predictions of our thermochemical modeling, which shows the cooling adiabat cross the graphite-liquid phase boundary at a similar time (Fig. 1a). The layered shell structure suggests the nano-onions form via surface-induced crystallization, leading the particles to solidify from the gas-liquid interface inwards[35]. We nonetheless consider a number of competing hypotheses. First, we address the possibility of a nanodiamond precursor. Conversion of nanodiamond to nano-onions has been observed by thermal vacuum treatment of detonation nanodiamond at T >~2000 K[28,36] on long time-scales; however, only trace, if any, amounts of nano-onions have ever been recovered from nanodiamond producing explosives, e.g. Comp B, where such a nanodiamond to nano-onion conversion scenario should also be operable[28]. Moreover, thermochemical modeling employing the experimentally observed nanocarbon particle size, ~9 nm diameter (Fig. 1a), suggests that the most probable DNTF expansion adiabat passes almost directly from the liquid to the graphite phase, precluding the formation of nano-diamonds as a major phase. The average diameter of the early-time carbon condensates (~9 nm) (Supplementary Fig. 4) and the collected nano-onions (~10 nm) (Supplementary Fig. 1) are much larger than typical detonation nanodiamond (4–5 nm)[2], which also suggests direct liquid phase formation and evolution. Direct gas phase nucleation and growth, similar to chemical vapor deposition, of nano-onions is also unlikely given the small change in the particle diameter size distribution observed over early times from the Guinier knee analysis (~9 nm to ~10 nm). Finally, the prompt solidification at t ~ 200 ns also limits the viability of a nucleation and growth model often invoked for detonation nanodiamond production[1,2], at least for high detonation temperature explosives such as BTF[37]. Further experimental work and/or simulations are needed to elucidate the early stages of liquid carbon droplet formation in these materials.

The consistency of our observations for recovered HNS and Comp B detonation products with reports in the literature give us confidence in our approach to understanding nano-onions pervasive throughout the recovered DNTF detonation products. The

TR-SAXS data, thermochemical modeling, and analysis of recovered solid detonation products converge on the conclusion that liquid droplets form rapidly behind the DNTF detonation front, and then freeze into graphitic carbon nano-onions. TR-SAXS reveals the initial formation of smooth, homogenous particles, consistent with liquid droplets, that evolve into a layered structure consistent with the nano-onions ubiquitous in the recovered detonation products of DNTF. The dynamics of this transition are observed in the TR-SAXS over the first few hundred nanoseconds post-detonation, and are remarkably consistent with thermochemical modeling of the particle transitioning from the liquid to graphitic phase at ~200 ns, post-detonation. Given the technological impact of detonation nanodiamonds, the efficient production of nano-onions in detonation could also have significant practical applications[38].

## Methods

**Materials**. Pressed cylindrical pellets of HNS, Comp B and DNTF were prepared at the High Explosives Applications Facility (HEAF) at LLNL. Cylinders were uniaxially pressed at ambient temperature from precursor powders. HNS IV and DNTF parts were pressed in a 6.35 mm diameter die at ambient temperature, 400 mg, to densities of about 1.56 g cm$^{-3}$ and 1.71 g cm$^{-3}$ respectively. Composition B (60% RDX and 40% TNT) 2 g parts were pressed in a 9.53 mm diameter die to 1.69 g cm$^{-3}$.

**In Operando Time-Resolved SAXS and Radiography**. Time-resolved SAXS measurements were performed within a LLNL detonation tank at the Dynamic Compression Sector located at the Advanced Photon Source (APS), Argonne National Laboratory, within the special purpose hutch (35-IDB)[21,24,29,30,39] or at 32-IDB[40]. Single pellets of HE were placed into a spring-loaded fixture with a piezo timing pin (Dynasen) positioned at the center, top of the sample. To detonate the HE, an exploding foil initiator (EFI) or an EFI-based detonator was placed underneath the pellet[40]. This assembly was placed within a ~120 L steel vacuum vessel (Teledyne RISI) and pumped down to < 200mTorr. The tank provides upstream and downstream Kapton$^{TM}$ (polyimide) windows to facilitate the X-ray transmission geometry required for SAXS and/or radiography. Within the vacuum vessel, Lexan$^{TM}$ (polycarbonate) panels ranging from 0.5 to 7 mm were placed ~200 mm from the sample in the beam path as shrapnel shields.

Detonation is synchronized with the APS bunch clock, thus permitting either SAXS or transmission radiography[41] from discrete 34 ps rms X-ray pulses, which arrive every 153.4 ns during 24-bunch mode. In the 1$^{st}$ Campaign of experiments, there was less certainty in the timing due to repeatability of sample motors, which were subsequently replaced. Therefore, the time certainty is based on the presence of partially detonated HE scattering in one of the frames in the TR-SAXS data (Supplementary Fig. 8). The time certainty of these experiments is estimated to be within ± 75 ns, associated with time between electron bunches. This uncertainty was not present in later experiments, where the time from detonation was certain to within ± 5 ns. Overall, a time resolution, governed by the detonation wave curvature, of < 50 ns was achieved by interleaving a series of shots with a small relative delay[21]. The sample-detector distance was about 1 m. Scattering intensity was recorded using an array of four identical area detectors (PI-MAX4 ICCD, Princeton Instruments) focused on the output of a scintillator and image intensifier[24,42–44]. The two-dimensional camera intensity images were reduced to one-dimensional SAXS traces using the Nika package[45] for the four cameras independently. Silver behenate and glassy carbon data acquired intermittently in the same vessel were used for calibration. Small-angle X-ray scattering data were subsequently reduced, intensity calibrated, and analyzed (Supplementary Eqs. 7–12; Supplementary Figs. 7, 8, 10, and 11) using the Irena package[46,47]. We note that the error bars Fig. 2a were calculated using the standard error of mean within Nika. For each point, the error is equal to the standard deviation divided by the square root of the number of pixels within the azimuthal integration. Further details of these procedures can be found in the supporting materials.

**Detonation product recovery and TEM charaterization**. Recovery of carbonaceous denotation products from the synchrotron-based denotation endstation was not feasible due to the necessary inclusion of several significant spectator sources of carbon, of which the primary contribution is acrylic plastics used to support and align the high explosive charge (comparable metal components would generate shrapnel that could penetrate the line-of-sight X-ray/vacuum windows). Consequently, three dedicated aluminum vessels were fabricated at HEAF for detonation of HNS, Comp B and DNTF, respectively, with relatively few sources of carbon contamination. The vessels were polished, cleaned with isopropanol and air-dried prior to use. Ice both quenches the products and provides a convenient medium to capture and retain detonation products. To form the ice, the interior volume of each aluminum vessel was filled with DI water. A Cu rod was immersed along the center axis. The entire assembly was placed in the freezer. Once the water was fully frozen, the Cu rod was heated and removed from the ice to create a bore along the center of the vessel, where the HE charge was subsequently placed. After detonation, the ice was removed and melted. The resulting suspension contained an abundance of carbonaceous detonation products. Aliquots of the recovered detonation products were taken from the collection vessels and further diluted in DI water. A few microliters of the diluted detonation product suspensions were drop cast on either an 8 nm thick silicon nitride TEM membrane (Ted Pella) or a lacey carbon TEM grid (Ted Pella). The sample droplets were left on the TEM grids for up to three minutes prior to wicking away the residual liquid with a microfiber cloth and drying with nitrogen gas. The samples thus prepared were kept in a TEM grid box under ambient conditions until analysis in the TEM.

TEM analysis was performed on a JEOL 2100 F field-emission microscope operating at either 120 kV or 200 kV. Images and diffraction patterns were acquired with an Orius SC1000 CCD at 1x binning, and EELS spectra were collected with a GIF Tridiem spectrometer (Gatan, Inc.) using a dispersion of 0.1 eV. Observable beam damage to the samples occurred only when the beam was left stationary for a prolonged time while operating in STEM mode. Substantial graphitization of nanodiamond or structural damage to the carbon onions was not observed during normal TEM imaging or collection of EELS spectra.

Each of the three recovered detonation products was characterized with the TEM multiple times (between ten and twenty microscope sessions). Multiple sample preparations were analyzed for each sample, and additionally single grids were examined multiple times over the course of many months post-preparation. No significant changes in the composition were observed. Many hundreds of TEM images, tens of diffraction patterns, and tens of EELS 1D and 2D maps were acquired and analyzed for each recovered detonation product. From these data sets, unique constituents of the detonation products were identified. A mixture of diamond and graphitic carbon, with a substantial fraction of diamond, was observed in the Comp B detonation products. A mix of the same two allotropes but with a much higher fraction of the graphitized ribbons was seen in HNS. In DNTF, no substantial contribution to the detonation product from any form of carbon aside from the nano-onions was observed (see, e.g., Supplementary Fig. 1).

**Thermochemical Modeling**. The size of the carbon condensates is traditionally employed as an empirical parameter in chemical equilibrium calculations of carbon-rich explosives by considering the carbon chemical potential corrections due to surface energy effects[15]. Results obtained for the detonation velocity of many energetic materials and the shock Hugoniots of various organic compounds suggest that the approach has clear practical value[7,13–15]. For the present study, we adopted a bootstrap computational methodology, where the average size of the carbon clusters is set to the early time value deduced from the SAXS experiments for the particular explosive. We calculated the carbon phase diagram using models[48] corresponding to each experimentally determined size by taking into account the surface energy contributions, and at the same time performed thermochemical calculations of detonation and expansion using for each material its specific condensate dimension. (For example, DNTF calculations included carbon (diamond, graphite, liquid), $N_2$, $CO_2$ and CO as major products, as well as minor gaseous products NO, $NO_2$, $N_2O$, $O_2$, etc., with total molar fraction less than 2%.) The reactivity of the condensed carbon, i.e., the extent to which it remains in chemical equilibrium with the gas products during expansion, is important for such estimates but it is difficult to ascertain due to the complexity of the processes involved; it is usually inferred from recovery and performance (e.g., energy delivery) experiments[13]. These are not available for DNTF, and we therefore assumed here that half of the excess DNTF carbon is likely to persist in particulate form on expansion and eventually be recovered, as suggested by experiments for a variety of energetic materials;[31] the reported calculation results and conclusions are robust with respect to sizable changes in the amount of recoverable carbon. With these assumptions the most probable (P,T) expansion paths for HNS, Comp B, and DNTF are shown in Fig. 1a. For the case of DNTF we also performed two-dimensional simulations of detonating charges with the exact size and density used in the experiments, coupling hydrodynamics and thermochemistry[49] with the goal of assessing the evolution of carbon clusters, initially predicted to be in the liquid phase, in the detonation shockwave flow. For these calculations we assumed that the freezing transition of the carbon particles to the local thermodynamically favored state, either diamond or graphite, is instantaneous on the time scale of the flows, and that diamond, once produced, does not further transform to graphite. The later assumption is in agreement with detonation recovery experiments for explosives such as Comp B, where nanodiamonds are by far the dominant phase. The simulations suggest that the liquid carbon clusters initially generated in the detonation should overwhelmingly transform to graphite upon expansion. They also enable an estimate of the timing of this transformation as observed by the SAXS beam traversing the expanding products—this is shown in Fig. 2 together with the experimental results.

## Data availability

All relevant data supporting the key findings of this study are available within the article and its Supplementary Information files or from the corresponding author upon reasonable request. The source data underlying Figs. 2, 3b, and 4d are provided as a Source Data file.

## Code availability

All codes used in this study are previously described in the cited literature.

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

## Acknowledgements

We thank and acknowledge B. Wood of LLNL, for providing calculated electron density maps across graphitic sheets. The authors also wish to acknowledge technical assistance from HEAF staff at LLNL including: D. Hansen, C. Mclean, S. Pease, K. Moua, B. Peralta, D. Voloshin, W. Bassett, N. Anderson, and F. Gagliardi. TR-SAXS experiments were further supported by the efforts of DCS staff including, D. Paskvan, T. Graber, and P. Rigg at WSU/DCS as well as additional technical input from T. Gog, S. Seifert, and J. Ilavsky at APS/ANL and J. Mang and D. Podlesak at LANL. This work was performed under the auspices of the U.S. Department of Energy by Lawrence Livermore National Laboratory under Contract DE-AC52–07NA27344 and was supported by the LLNL-LDRD Program under Project No. 14-ERD-018. MHN acknowledges support from the Lawrence Fellowship. Work at the Molecular Foundry was supported by the Office of Science, Office of Basic Energy Sciences, of the U.S. Department of Energy under Contract No. DE-AC02–05CH11231. The Dynamic Compression Sector at the Advanced Photon Source is managed by Washington State University and funded by the National Nuclear Security Administration of the U.S. Department of Energy under Cooperative Agreement No. DE-NA0002442. Supporting experiments and data were also performed at 32-ID-B at APS. This research used resources of the Advanced Photon Source, a U.S. Department of Energy (DOE) Office of Science User Facility operated for the DOE Office of Science by Argonne National Laboratory under Contract No. DE-AC02–06CH11357.

## Author contributions

T.M.W., T.V.B., L.E.F. and S.B. conceived the project. T.M.W., M.B.H and T.V.B, designed the TR-SAXS experiments. M.B.H., J.A.H., M.H.N., L.M.L., R.L.H., A.J., W.L.S., E.V.B., N.S., R.L.G., E.B.W., B.J.J., D.M.D., M.A.F., R.C.H., B.S.R., J.R.I.L., T.V.B. and T.M.W. assembled experimental systems and collected TR-SAXS data at the APS. L.M.L., R.L.H. and C.M. designed all detonation fixturing, coordinated HE logistics, assembled HE shots, and led firing operations. S.B. and L.E.F. designed and executed thermochemical modeling. P.P. synthesized DNTF. M.B.H., L.M.L., M.H.N

and T.M.W designed and conducted post-detonation recovery experiments in ice. M.H.N. and S.A. collected TEM data. J.A.H. carried out TR-SAXS modeling. The paper was written by M.B.H., J.A.H., M.H.N., S.B. and T.M.W. with contributions and feedback from all authors.

## Additional information

**Competing interests:** The authors declare no competing interests.

