## [Peer Review File · Nature Communications]

Reviewers' comments:

Reviewer #1 (Remarks to the Author):

Bagge-Hansen et al. present an interesting study of in situ small angle X-ray scattering studies on detonation waves, presumably showing the formation of carbon nano onions for DNTF, nanodiamonds for composition B and a mixture of different carbon nano particles for HNS. The study seems to be mostly well done and technically sound. I recommend publication after addressing several suggestions for changes/additions.

1. Much of the involved physics is hidden in the supplementary material. I think showing more details in the main article would be well in line with the guidelines of Nature Communications. In particular, I recommend discussing the fitting model for the nano onion data in the main article (maybe not in all detail, but at least introducing the relevant equations). Moreover, it would be good to show a few exemplary fits as well as the size distributions inferred from SAXS in the main article.
2. I wonder why the synchronization accuracy with the synchrotron is only ~50 ns. It should be possible to do way better on synchrotrons. See e.g. R. Torchio et al., *Sci Rep.* 6, 26402 (2016) or M. P. Olbinado et al., *J. Phys. D: Appl. Phys.* 51, 055601 (2018). What is the problem in the discussed experiment? Moreover, the temporal error bars in figure 2b are even larger than the claimed 50 ns. The reason does not seem clear. Furthermore, it seems strange or unlucky, that no intermediate state for $\Delta\rho$ was observed. The very interesting time window between 100 ns and 200 ns seems to have a lack of data points. Was this on purpose or just unlucky?
3. To get an impression of the SAXS signal of DNTF at ambient conditions and potential influence on the signal interpretation, I recommend adding a lineout for ambient conditions ($t = 0$ ns) to figure 2a. So far, only Figure S8 discusses this a little bit.
4. Figure 3: The absolute scale of the lineouts seems to be inconsistent with what is shown in the supplemental material, e.g. Figure 3: maximum signal of composition B is ~700, whereas it is ~100 in Figure S11.
5. I recommend adding a detector raw data image, at least to the supplement. This will give a better idea on the clarity of the data and also help with clarifying the different steps to obtain the final lineouts.

6. The authors may want to discuss how such experiments may benefit in the future from simultaneous X-ray diffraction and SAXS, as now enabled by X-ray free electron lasers, see. e.g. D. Kraus et al., Phys. Plasmas 25, 056313 (2018), and potentially also the upcoming next generation of synchrotrons. I assume that adding diffraction will help a lot with identifying graphite or diamond structures inside the carbon nano particles.

7. Why do the authors use a different model for fitting the data from composition B? And why actually assuming monodisperse particles in this case? It would be a very nice cross-check to see whether the composition B data can be fitted with the nano onion model with very low $\Delta\rho$.

8. From the signal intensity, the authors claim to observe volume fractions of the carbon nano particles up to 10%. Did the authors check, if their model, which is based on the summation of single dilute nano particles, still holds in this regime of relatively densely packed nano particles?

9. In the supplemental material, the equation references shown in figure S8 and figure S12 seem to point to the wrong equations.

10. Figure S5, middle left pane: the fit for 0.055 μs seems to be very bad. I would assume this can be done better.

Reviewer #2 (Remarks to the Author):

Dr. Willey and co-workers have probed the early carbon condensation kinetics occurring during detonation of high explosives, using time-resolved small-angle X-ray scattering.

Complementary studies with transmission electron microscopy and electron energy loss spectroscopy on the final products of the reactions allowed fixing some of the variable values, making the scattering data analysis more robust. Finally, numerical simulations of the probed systems validate the proposed interpretation.

The study, based on a challenging experiment, is relevant for the unprecedented time resolution attained, better than 50 ns. It is fundamental to understand the early stages of the reactions and, therefore, better control them.

The small-angle X-ray scattering data analysis is discussed using strong evidence.

Individual queries:

1) Line 84-85 of main text: “the undetonated DNTF manifests as scattering intensity at low- q , where q is the modulus of the scattering vector, associated with micron-scale porosity in the pressed pellet.”

Please, add a reference to Figure S8 of Supplementary Material.

2) Figure 3: Small-angle X-ray scattering profiles at $t=0$ can be added to show how the kinetics evolves.

Line 84-85 of the manuscript can then refer to this figure.

Is there a relationship of the phase diagram with the scattering curves at $t=0$?

3) Line 62 of Supplementary Material: The sentence is not clear.

4) Equation 2 of Supplementary Material: Please, define p_c and p_d .

5) Figure S9 of Supplementary Material: Please, add notation "a" and "b" to the two graphs and use them on the figure caption.

Reviewer #3 (Remarks to the Author):

In general, it is an interesting and important work since mechanisms of nanodiamond formation from explosives are still debated. The authors mainly observed formation of carbon nanoions and explain it as a result of direct liquid carbon to nanoion transition for a particular explosive DNTF, which is interesting.

Do the authors think that the preferential formation of carbon nanoions from this particular explosive is related to the absence of water in detonation products in this case? In particular, due to the absence of water cooling effect (which is confirmed by the highest predicted detonation temperature for DNTF), which may promote carbon graphitization versus diamond formation?

There are several other sticking points in this work that need to be taken care of before the manuscript could be reconsidered for publication again (please find comments embedded in the attached .pdf)

Detonation synthesis of carbon nano-onions via liquid carbon condensation

M. Bagge-Hansen¹, S. Bastea¹, J. A. Hammons¹, M. H. Nielsen¹, L. M. Lauderbach¹, R. L. Hodgkin¹, P. Pagoria¹, C. May¹, S. Aloni², A. Jones¹, W. L. Shaw¹, E. V. Bukovsky¹, N. Sinclair³, R. L. Gustavsen⁴, E. B. Watkins⁴, B.J. Jensen⁴, D. M. Dattelbaum⁴, M. A. Firestone⁴, R. C. Huber⁴, B. S. Ringstrand⁴, J. R. I. Lee¹, T. van Buuren¹, L. E. Fried¹, and T. M. Willey¹

Format:

Reviewer Comment.

Author Replies.

Reviewer 1:

Bagge-Hansen et al. present an interesting study of in situ small angle X-ray scattering studies on detonation waves, presumably showing the formation of carbon nano onions for DNTF, nanodiamonds for composition B and a mixture of different carbon nano particles for HNS. The study seems to be mostly well done and technically sound. I recommend publication after addressing several suggestions for changes/additions.

The authors would like to thank the reviewer for a thoughtful consideration of this work. We have carefully considered all of the reviewer's remarks and the manuscript is improved as a result. We hope that the changes and discussion below satisfactorily address the reviewer's concerns.

1. Much of the involved physics is hidden in the supplementary material. I think showing more details in the main article would be well in line with the guidelines of Nature Communications. In particular, I recommend discussing the fitting model for the nano onion data in the main article (maybe not in all detail, but at least introducing the relevant equations). Moreover, it would be good to show a few exemplary fits as well as the size distributions inferred from SAXS in the main article.

This point is well made. The SAXS modeling was discussed almost entirely in the supplementary information in order to keep the article as concise as possible, but we have now included an abbreviated discussion of the SAXS modeling behind Figure 2 in the main text. In order to maintain some brevity, the model derivation from the spherically symmetric phase is explained with references to Discussion, Figures and Equations in the supplementary information. The analytical solution to the well-known scattering equation for a spherically symmetric phase with a Heaviside radial distribution function can be found in the supplemental information along with the incorporation of the size distribution, volume fraction and flat background. We hope the reviewer agrees that this summary is sufficient for a person with competence in small angle

scattering to derive the rest of the equations and a person with no small angle scattering experience to understand the general approach.

The model fits are now included in Figure 2a along with the TR-SAXS from the static DNTF pellet prior to detonation (comment below). We considered the addition of the size distributions obtained from the TR-SAXS to the main article. Ultimately, only the mean value is of import to the discussion and we opted to draw the interested reader to the supplementary information.

2. I wonder why the synchronization accuracy with the synchrotron is only ~50 ns. It should be possible to do way better on synchrotrons. See e.g. R. Torchio et al., Sci Rep. 6, 26402 (2016) or M. P. Olbinado et al., J. Phys. D: Appl. Phys. 51, 055601 (2018). What is the problem in the discussed experiment? Moreover, the temporal error bars in figure 2b are even larger than the claimed 50 ns. The reason does not seem clear. Furthermore, it seems strange or unlucky, that no intermediate state for $\Delta\rho$ was observed. The very interesting time window between 100 ns and 200 ns seems to have a lack of data points. Was this on purpose or just unlucky?

The time uncertainty is determined by a number of factors, such as uncertainty in the relative position of the X-ray beam from the top of the HE pellet. In the first Campaign, there was concern that the motor used to adjust the pellet height was not behaving within specification. Therefore, the time uncertainty of Campaign 1 (+/- 75 ns) was higher than in Campaign 2 (+/- 5 ns) after the motor was replaced. Text explaining this has been included in the second part of the Methods section.

In Figure 2b, we were in fact unlucky with timing in one interleaved shot in our second experimental campaign (campaign 2, experiment 4 in Fig 2b) and did not collect an additional point at ~125 ns due to an abnormal detonator function time; however, experiment 4 was fortuitously nearly synchronized with experiment 3 and demonstrates the good shot-to-shot repeatability. DNTF is relatively rare and difficult to synthesize, and we were unable to field additional shots.

Further, the intensity variation between $5 \times 10^{10} \text{ cm}^{-2}$ and $15 \times 10^{10} \text{ cm}^{-2}$ in Fig. 2b is abrupt in the transitional region (100 ns to 200 ns) because of the highly nonlinear dependence of the SAXS data on $\Delta\rho$, high (flat) background and low signal-to-noise. In the supplemental information, this non-linearity is illustrated in Figure S3. The data quality of the TR-SAXS is not sufficient to reliably resolve a contrast for transitional states between $5 \times 10^{10} \text{ cm}^{-2}$ and $15 \times 10^{10} \text{ cm}^{-2}$. In most all cases, it is not possible for least squares fitting to determine an optimal value of $\Delta\rho$ in the range: $5 \times 10^{10} \text{ cm}^{-2} < \Delta\rho < 15 \times 10^{10} \text{ cm}^{-2}$ because there is no minimum for χ^2 in this range, and we have reflected this in the error bars for these points. We have also edited the main text to further explain the non-linear behavior of $\Delta\rho$ and the importance of this behavior to Fig. 2b.

3. To get an impression of the SAXS signal of DTNF at ambient conditions and potential influence on the signal interpretation, I recommend adding a lineout for ambient conditions ($t = 0$ ns) to figure 2a. So far, only Figure S8 discusses this a little bit.

We thank the reviewer for this improvement to the clarity of Fig 2a. In all of the experiments, the SAXS from the static HE pellet was obtained prior to detonation. The DNTF ($t < 0$ ns) SAXS

traces (blue) are now included in Figure 2a. $t = 0$ ns is defined as the moment the detonation front arrives, along the center axis of the cylindrical pellet, at the elevation of the X-ray beam.

4. *Figure 3: The absolute scale of the lineouts seems to be inconsistent with what is shown in the supplemental material, e.g. Figure 3: maximum signal of composition B is ~ 700 , whereas it is ~ 100 in Figure S11.*

Upon investigation, we discovered that the Composition B data in Figure 3 were not scaled by the calibration factor obtained from glassy carbon. The calibration factor was applied to the data and Figure 3 was updated.

5. *I recommend adding a detector raw data image, at least to the supplement. This will give a better idea on the clarity of the data and also help with clarifying the different steps to obtain the final lineouts.*

The raw TR-SAXS images obtained from the experiment from Campaign 1 shown in Figure 2a were included in the Data Reduction section of the Supplementary Information document. In addition to the actual raw images, the dark field, empty chamber and Ag behenate (q-calibrant) were also included.

6. *The authors may want to discuss how such experiments may benefit in the future from simultaneous X-ray diffraction and SAXS, as now enabled by X-ray free electron lasers, see. e.g. D. Kraus et al., Phys. Plasmas 25, 056313 (2018), and potentially also the upcoming next generation of synchrotrons. I assume that adding diffraction will help a lot with identifying graphite or diamond structures inside the carbon nano particles.*

The main advantage of free electron lasers is a much higher signal-to-noise via higher photons per pulse, and much higher monochromaticity. Ultimately, better time resolution would provide much more information but is limited by X-ray detection cameras as well as the detonation wave curvature. Indeed, simultaneous diffraction would be extremely valuable and is actively being explored by this team in a series of time-resolved WAXS experimental campaigns.

7. *Why do the authors use a different model for fitting the data from composition B? And why actually assuming monodisperse particles in this case? It would be a very nice cross-check to see whether the composition B data can be fitted with the nano onion model with very low $\delta\rho$.*

The main difference between composition B and DNTF is the high-q scattering where the scattered intensity transitions from a Porod decay to a shallower decay. While there is observable high-q scattering in composition B, it is modeled using a second feature size in the two-level unified equation (another Guinier knee). It is worth noting that poor model fits are obtained when the scattering from composition B is modeled in the same way as the late-time DNTF because the high-q scattering is not the same. The approach to scattering analysis for

composition B is somewhat controversial and is the topic of a draft manuscript that is currently under review; consequently, a simple the two-level unified equation is fit to composition B. This model assumes a size distribution with a log-normal standard deviation of ~ 0.3 , which is consistent with TEM imaging.

The reviewer is correct that a size distribution of low $\Delta\rho$ onions (spheres) can be fit to the data using, for example, regularization, method of maximum entropy or the interior point gradient method. However, such a fit does not provide any additional information than the unified equation and there is no evidence (either from thermochemical simulations or TEM imaging) that carbon onions form during a composition B detonation.

8. From the signal intensity, the authors claim to observe volume fractions of the carbon nano particles up to 10%. Did the authors check, if their model, which is based on the summation of single dilute nano particles, still holds in this regime of relatively densely packed nano particles?

A precise volume fraction was not originally extracted from the SAXS modeling because the irradiated volume through the X-ray beam, transmission and scattering length density of detonation products (bulk and onions) are all changing slightly after the detonation wave passes. If one were to assume these changes were minimal and that the scattering length densities of the onions are exact, the volume fraction of the onions obtained from the modeling is slightly less than 1 %, which is safely below 10 % stated. Neither a broad peak nor “flat” region preceding the Guinier knee was observed in any of the TR-SAXS data, which would be present if there were particle correlations. Therefore, 10 % is errantly high. Rather than reporting a broad range, the volume fraction of onions was estimated with thermochemical simulations and the text in the supplementary information was changed to:

“Using Equation S6 and assuming a constant transmission, X-ray path length and scattering length density of the bulk, we estimate that the onions occupy less than 1 % of the volume fraction within 0.3 μs after the detonation wave passes. This number is very close to 2%, which is estimated from thermochemical simulation.”

9. In the supplemental material, the equation references shown in figure S8 and figure S12 seem to point to the wrong equations. The equation references in Figure S11 and Figure S12 were changed to 10. Figure S5, middle left pane: the fit for 0.055 μs seems to be very bad. I would assume this can be done better.

In general, adjusting the starting conditions or adding more fit parameters (e.g. inter-layer spacing) can improve all of the model fits. However, all of the TR-SAXS data was fit using consistent starting conditions and with as few fit parameters as possible. The model fit pointed out by the reviewer was improved by slightly adjusting the starting conditions for the size distribution. Figure S5 was updated accordingly.

Reviewer 2:

Dr. Willey and co-workers have probed the early carbon condensation kinetics occurring during detonation of high explosives, using time-resolved small-angle X-ray scattering.

Complementary studies with transmission electron microscopy and electron energy loss spectroscopy on the final products of the reactions allowed fixing some of the variable values, making the scattering data analysis more robust. Finally, numerical simulations of the probed systems validate the proposed interpretation.

The study, based on a challenging experiment, is relevant for the unprecedented time resolution attained, better than 50 ns. It is fundamental to understand the early stages of the reactions and, therefore, better control them.

The small-angle X-ray scattering data analysis is discussed using strong evidence.

The authors would like to thank the reviewer for a thoughtful consideration of this work, and the concise and positive summary provided above. We have carefully considered all of the reviewer's remarks and the manuscript is improved as a result. We hope that the changes and discussion below satisfactorily address the reviewer's concerns.

Individual queries:

1) Line 84-85 of main text: "the undetonated DNTF manifests as scattering intensity at low- q , where q is the modulus of the scattering vector, associated with micron-scale porosity in the pressed pellet."

Please, add a reference to Figure S8 of Supplementary Material.

A reference to the undetonated data in Figure 2a is now included in the main text.

2) Figure 3: Small-angle X-ray scattering profiles at $t=0$ can be added to show how the kinetics evolves.

Line 84-85 of the manuscript can then refer to this figure.

Is there a relationship of the phase diagram with the scattering curves at $t=0$?

The static DNTF ($t < 0$ ns) SAXS traces (blue) are now included in Figure 2a. $t = 0$ ns is defined as the moment the detonation front arrives, along the center axis of the cylindrical pellet, at the elevation of the X-ray beam. As such, the TR-SAXS data at $t=0$ is nearly entirely undetonated DNTF and has no relationship to the carbon phase diagram in Fig. 1a. That said, the pressures and temperatures of detonation, indicated approximately by the initial point of the cooling adiabat (Fig. 1a), are of great importance to the phase of the carbon condensates that form promptly after the detonation front passes. The evidence presented for liquid phase carbon condensates presented in this manuscript underscores this point.

3) Line 62 of Supplementary Material: The sentence is not clear.

There was some missing text in the original sentence. This sentence was intended to emphasize why ρ_{DNTF} was assumed to be that of the bulk, undetonated high explosive pellet. It was edited to read:

“Beyond the onion particle, the scattering length density of the bulk is approximated as undetonated DNTF, ρ_{DNTF} , since the mass has not expanded appreciably at early times that are important in this study.”

4) Equation 2 of Supplementary Material: Please, define ρ_c and ρ_d .

The text following Equation 2 was changed to:

“where, l represents the distance over steps where $\rho_{\text{onion}}(r)$ is held constant (half the graphitic layer to layer distance), ρ_c is the scattering length density of the electron-rich graphitic layer, ρ_d is the scattering length density of the electron-poor region between the graphitic layers, and the summation begins at l , ends at $R-l$ and increments by $2l$. In Equation S2, the function, $A_s(r, q, [\rho_c - \rho_d])$ is the well-known q -dependent scattering amplitude of a sphere with radius, r , and contrast, $\rho_c - \rho_d$ and $V(R)$ is the volume of a sphere.”

5) Figure S9 of Supplementary Material: Please, add notation “a” and “b” to the two graphs and use them on the figure caption.

The two plots in Figure S9 (now Figure S10) were noted as “a” and “b”. The Figure caption was changed to:

“Figure S10: (a) a plot of the silver behenate collected from a monochromatic X-ray beam at 9-IDC (black) with the expected scattering from the U18 with Rh mirrors (blue), and the measured scattering from the U18 with Rh mirror from the four different cameras (circles). The expected U18 undulator profile is shown in (b).”

Reviewer 3:

In general, it is an interesting and important work since mechanisms of nanodiamond formation from explosives are still debated. The authors mainly observed formation of carbon nanoions and explain it as a result of direct liquid carbon to nanoion transition for a particular explosive DNTF, which is interesting.

The authors would like to thank the reviewer for a thoughtful consideration of this work. We have carefully considered all of the reviewer’s remarks and the manuscript is improved as a

result. We hope that the changes and discussion below satisfactorily address the reviewer's concerns.

Do the authors think that the preferential formation of carbon nanoonions from this particular explosive is related to the absence of water in detonation products in this case? In particular, due to the absence of water cooling effect (which is confirmed by the highest predicted detonation temperature for DNTF), which may promote carbon graphitization versus diamond formation?

Water plays an important role in the detonation products of CHNO explosives, acting as a catalyst of high pressure and temperature reactions and thus facilitating the release of chemical energy [14]. Conversely, in hydrogen-free explosives the absence of water can slow down or even inhibit exothermic chemical processes, leading to faster cooling and decompression on expansion. In the case of DNTF this results in graphitization being favored over diamond formation, and the production of carbon nano-onions.

Added reference:[14] Christine J. Wu, Laurence E. Fried, Lin H. Yang, Nir Goldman and Sorin Bastea, Catalytic behavior of dense hot water, Nat. Chem. 1, 57 (2009).

There are several other sticking points in this work that need to be taken care of before the manuscript could be reconsidered for publication again (please find comments embedded in the attached .pdf)

The embedded comments and responses are extracted below:

Page 1, line 35

The sentence was edited to read: "..., e.g., in enabling more efficient production of other potentially useful carbon nanoallotropes, predicting the energy release behavior, identifying failure mechanisms, and decreasing sensitivity of HEs [5]; ..."

line 45

The sentence was edited to read: "..., which are a complex mixture of molecular gases (e.g. N₂, H₂O, CO, CO₂, etc.), ionic species (e.g. OH⁻, H⁺, etc.) and carbon condensates, evolve and reach full chemical equilibrium at the Chapman-Jouguet (C-J) point [11,12,13,14]."

Page 2, Line 85:

The sentence was edited to read: "..., associated with micron-scale features (e.g., voids) in the pressed pellet..."

Page 3, Lines 100-102:

This sentence was edited and relocated earlier in text. It now reads: "The interlayer spacing between concentric spherical shells is set to 3.1 Å, based on TEM results, below." This text is placed within discussion of TRSAXS modeling parameters. The reviewer's suggestion to contextualize the observation of a 3.1 Å interlayer spacing in the existing nano-onion literature

is a well-made. We have done this in the TEM section, where the interlayer spacing measurement is extracted from the microscopy data. It should be emphasized that the TRSAXS conclusions are not sensitive to the magnitude of the interlayer spacing.

Line 124: (now reorganized into discussion section)

The sentence was edited to read: “. The average diameter of the early-time carbon condensates (~9 nm) (supplementary figure 4) and the collected nano-onions (~10 nm) (supplementary figure 1) are much larger than typical detonation nanodiamond (4-5 nm)[2], which also suggests direct liquid phase formation and evolution.”

Lines 127-128

The sentence was edited to read: “Finally, the prompt solidification at $t \sim 200$ ns also limits the viability of a nucleation and growth model often invoked for detonation nanodiamond production [1,2], at least for high detonation temperature explosives such as BTF [37]. Further experimental work and/or simulations are needed to elucidate the early stages of liquid carbon droplet formation in these materials.”

Added reference: [37] Stepan S. Batsanov et al., Novel synthesis and properties of hydrogen-free detonation nanodiamond, *Mat. Chem. Phys.* 216, 120 (2018).

Page 6, lines 272-273:

Q: This contradicts many other literature sources where substantial nanodiamond graphitization was observed at these high accelerating voltages. The authors need to repeat their TEM studies at voltages below 100 eV (i.e., 80 eV)

There is a pervasive perception that one can readily convert between nanodiamonds and nano-onions using TEM irradiation. While there are reports in the literature of such conversions, we did not observe this phenomenon during TEM characterization of the recovered detonation products. Imaging and diffraction data were collected well within a minute of exposure to the electron beam, and no substantial damage to or change in the sample was observed during that time. After prolonged STEM/EELS characterization with long dwell times, structural damage to the various carbon allotropes was observed, as well as limited graphitization of nanodiamonds. However, total or majority conversion from one allotrope to another was never observed.

Transition	Temperature	Voltage (kV)	Dose (A/cm ²)	Time	References
Onion to ND	500 - 700 C	not in TEM	not in TEM	0.5 - 3 hours	1, 2
(D)ND to Onion	850 - 1700 C	not in TEM	not in TEM	5 - 90 minutes	3, 4, 5, 6, 7, 8, 9, 10
graphite to diamond	500 - 1300 K	400, 1250	5-240	> 2 hours	11, 12

Onions to ND	675 - 2275 K	200(?), 1250	10-200	15 minutes - hours	13, 14, 15, 16
(N)D to Onion	RT	200, 300	~1-150	5 (pre-treat) - 60 min	17, 18, 19
DND: some graphite	RT	200	~10-15	35 min	This work
Onion -> no diamond	RT	200	~10-15	35 min	This work

Table R1: Transformations between carbon onions and diamond as reported in selected literature. Data reported in references 1 – 10 cover transformation between carbon allotropes achieved by other methods with subsequent TEM characterization and no mention of allotrope interconversion under electron irradiation.

There is a substantial body of literature that utilizes TEM to characterize nanocarbon allotropes. A minute selection from this corpus, comprised of papers discussing allotrope transformation by other means can be found in the first two rows of Table R1 along with the time and temperature ranges necessary to achieve these transformations. The next two rows represent reports of in situ TEM observations of allotrope conversion. Conditions necessary to realize these events were at a minimum temperatures elevated by hundreds of degrees and times on tens to hundreds of minutes. Furthermore, in all but at most one instance (TEM operating conditions were not universally well-reported), higher accelerating voltages than are typically used were necessary. There were three reports that we found of TEM-induced transformation of diamond to nano-onion at room temperature, one on bulk diamond and two on nanodiamond. With a chemical pre-treatment, substantial onion formation was observed on nanodiamond after about five minutes in one report. In the absence of a pre-treatment, similar transformation was observed after 30 minutes.

Figure R1. Effect of TEM irradiation on carbon nano-onions found in recovered DNTF soot. After 35 minutes of continuous irradiation extensive defects in the graphite structure are observed, but original structure is still widely intact. FFTs from selected frames show persistence of graphitic layers throughout entire period of irradiation. Scale bar is 10 nm.

Figure R2. Effect of TEM irradiation on detonation nanodiamonds found in recovered Composition B soot. After 35 minutes of continuous irradiation limited graphitization of the nanodiamond surface is observed, but the original particles are still widely intact. FFTs from selected frames show persistence of diamond structure throughout entire period of irradiation. In order to verify that the structures reported in the manuscript are native to the recovered detonation products and are not produced during TEM characterization, we collected in situ TEM videos of the carbonaceous detonation products irradiated at similar beam conditions to those used to collect the data in Figure 4. Video was collected at a rate of one frame per second for 35 minutes, after which static images were collected as well as diffraction and core edge EELS data. Selected frames from the videos and their FFTs are shown above in Figures R1 and R2 for the recovered DNTF and Composition B detonation products, respectively. While structural damage to the onion graphite layers and graphitization of the nanodiamond surface are evident, substantial conversion from one allotrope to another was not observed. The 002 band in the onion FFT broadens out at later times as the graphite structure is progressively damaged, but there's no indication of diamond formation. At the latest time for the nanodiamond FFT there's a slight, and very broad, increase in intensity around the graphite 002 region, but the FFT still shows a strong signal from nanodiamond. Scale bar is 10 nm.

The report in the literature with the most similar TEM operating conditions to those used in the current work are in Ref. 19 where Roddatis *et al.* observed onion formation from nanodiamond within a few minutes using the same accelerating voltage and a similar beam dose to what we used here. The obvious difference between conditions is the chemical wash that they used on the nanodiamonds prior to TEM characterization. Perhaps this pre-treatment rendered the nanodiamonds more susceptible to transformation to nano-onions under TEM irradiation.

Instead of "(These calculations included ..."
we have replaced with the text: "(For example, DNTF calculations included ..."

page 7 lines 316-317

Q: Is this conclusion valid for all considered HEs or only DNTF?

This conclusion applies to DNTF, consistent with its expansion path shown in Fig. 1a.

References:

1. Tomita S, Fujii M, Hayashi S, Yamamoto K. Transformation of carbon onions to diamond by low-temperature heat treatment in air. *Diam Relat Mater* **9**, 856-860 (2000).
2. Sun J, Xu F, Sun LT. In situ investigation of the mechanical properties of nanomaterials by transmission electron microscopy. *Acta Mech Sinica-Prc* **28**, 1513-1527 (2012).
3. Kuznetsov VL, Chuvilin AL, Butenko YV, Malkov IY, Titov VM. Onion-Like Carbon from Ultra-Disperse Diamond. *Chemical Physics Letters* **222**, 343-348 (1994).
4. Mykhailiv O, *et al.* Influence of the Synthetic Conditions on the Structural and Electrochemical Properties of Carbon Nano-Onions. *Chemphyschem* **16**, 2182-2191 (2015).
5. Mykhaylyk OO, Solonin YM, Batchelder DN, Brydson R. Transformation of nanodiamond into carbon onions: A comparative study by high-resolution transmission electron microscopy, electron energy-loss spectroscopy, x-ray diffraction, small-angle x-ray scattering, and ultraviolet Raman spectroscopy. *J Appl Phys* **97**, (2005).
6. Qiao ZJ, Li JJ, Zhao NQ, Shi CS, Nash P. Graphitization and microstructure transformation of nanodiamond to onion-like carbon. *Scripta Mater* **54**, 225-229 (2006).
7. Dubitsky GA, *et al.* Effect of high pressures and temperatures on carbon nano-onion structures: comparison with C-60. *Russ Chem B+* **60**, 413-418 (2011).
8. Joly-Pottuz L, *et al.* Diamond-derived carbon onions as lubricant additives. *Tribol Int* **41**, 69-78 (2008).
9. Zeiger M, Jackel N, Weingarth D, Presser V. Vacuum or flowing argon: What is the best synthesis atmosphere for nanodiamond-derived carbon onions for supercapacitor electrodes? *Carbon* **94**, 507-517 (2015).

10. Zou Q, Wang MZ, Li YG, Lv B, Zhao YC. HRTEM and Raman characterisation of the onion-like carbon synthesised by annealing detonation nanodiamond at lower temperature and vacuum. *J Exp Nanosci* **5**, 473-487 (2010).
11. Lyutovich Y, Banhart F. Low-pressure transformation of graphite to diamond under irradiation. *Appl Phys Lett* **74**, 659-660 (1999).
12. Zaiser M, Lyutovich Y, Banhart F. Irradiation-induced transformation of graphite to diamond: A quantitative study. *Phys Rev B* **62**, 3058-3064 (2000).
13. Banhart F. The transformation of graphitic onions to diamond under electron irradiation. *J Appl Phys* **81**, 3440-3445 (1997).
14. Huang JY. In situ observation of quasimelting of diamond and reversible graphite-diamond phase transformations. *Nano Lett* **7**, 2335-2340 (2007).
15. Redlich P, Banhart F, Lyutovich Y, Ajayan PM. EELS study of the irradiation-induced compression of carbon onions and their transformation to diamond. *Eur Mat Res* **68**, 561-563 (1998).
16. Zaiser M, Banhart F. Radiation-induced transformation of graphite to diamond. *Physical Review Letters* **79**, 3680-3683 (1997).
17. Hiraki J, Mori H, Taguchi E, Yasuda H, Kinoshita H, Ohmae N. Transformation of diamond nanoparticles into onion-like carbon by electron irradiation studied directly inside an ultrahigh-vacuum transmission electron microscope. *Appl Phys Lett* **86**, (2005).
18. Qin LC, Iijima S. Onion-like graphitic particles produced from diamond. *Chemical Physics Letters* **262**, 252-258 (1996).
19. Roddatis VV, Kuznetsov VL, Butenko YV, Su DS, Schlogl R. Transformation of diamond nanoparticles into carbon onions under electron irradiation. *Phys Chem Chem Phys* **4**, 1964-1967 (2002).

REVIEWERS' COMMENTS:

Reviewer #1 (Remarks to the Author):

The authors have responded satisfactory to all my comments and thus, I recommend publication.

Reviewer #2 (Remarks to the Author):

Dr. Willey and co-authors report on studies of early carbon condensation kinetics during explosives detonation.

The referee comments have been addressed appropriately, resulting in an improvement of the manuscript.

Individual query

Line 109: please add a reference to the scattering equation

Reviewer #3 (Remarks to the Author):

This reviewer believes that the authors have sufficiently addressed concerns by the reviewers in their revised manuscript and hence, it is recommended for publication now.